# Enhanced UV-B Radiation Induced the Proanthocyanidins Accumulation in Red Rice Grain of Traditional Rice Cultivars and Increased Antioxidant Capacity in Aging Mice

**DOI:** 10.3390/ijms24043397

**Published:** 2023-02-08

**Authors:** Xiang Li, Jianjun Sheng, Zuran Li, Yongmei He, Yanqun Zu, Yuan Li

**Affiliations:** 1College of Resources and Environment, Yunnan Agricultural University, Kunming 650201, China; 2National Engineering Research Center for Ornamental Horticulture, Flower Research Institute, Yunnan Academy of Agricultural Sciences, Kunming 650231, China; 3College of Horticulture and Landscape, Yunnan Agricultural University, Kunming 650201, China

**Keywords:** UV-B radiation, red rice, proanthocyanidins, distribution, synthesis, antioxidant capacity

## Abstract

Proanthocyanidins are major UV-absorbing compounds. To clarify the effect of enhanced UV-B radiation on the proanthocyanidin synthesis and antioxidant capacity of traditional rice varieties in Yuanyang terraced fields, we studied the effects of enhanced UV-B radiation (0, 2.5, 5.0, 7.5 kJ·m^−2^·d^−1^) on the rice grain morphology, proanthocyanidins content, and synthesis. The effects of UV-B radiation on the antioxidant capacity of rice were evaluated by feeding aging model mice. The results showed that UV-B radiation significantly affected the grain morphology of red rice and increased the compactness of starch grains in the starch storage cells of central endosperm. The content of proanthocyanidin B2 and C1 in the grains was significantly increased by 2.5 and 5.0 kJ·m^−2^·d^−1^ UV-B radiation. The activity of leucoanthocyanidin reductase was higher in rice treated by 5.0 kJ·m^−2^·d^−1^ than other treatments. The number of neurons in the hippocampus CA1 of mice brain fed red rice increased. After 5.0 kJ·m^−2^·d^−1^ treatment, red rice has the best antioxidant effect on aging model mice. UV-B radiation induces the synthesis of rice proanthocyanidins B2 and C1, and the antioxidant capacity of rice is related to the content of proanthocyanidins.

## 1. Introduction

The enhancement of surface ultraviolet (UV-B, 280–315 nm) radiation caused by ozone attenuation is one of the major global climate changes [1]. In recent years, the ozone-depleting substance emission has increased. The ozone in the lower stratosphere in the densely populated low-latitude region of the earth is still decreasing [2]. With the increase in the surface UV-B radiation intensity, the response of various elements in the ecosystem to UV-B radiation has become a research focus. After receiving UV-B radiation photons, plant UVR8 photoreceptors act as signal regulators and activate defense mechanisms through gene transcription [3]. The accumulation of secondary metabolites is a marker process of the plant response to UV-B radiation, such as phenolic compounds (especially flavonoids), alkaloids, and terpenoids. Flavonoids mainly include anthocyanins, flavonols, flavanols, and proanthocyanidins. They can greatly absorb UV-B radiation and basically do not absorb visible light. They are a major UV-B radiation shielding material [4]. Therefore, the impact and mechanism of enhanced UV-B radiation on the metabolism and synthesis of plant flavonoids have received extensive attention, but this has mainly been focused on the impact of enhanced UV-B radiation on the content and synthesis of flavonoids in plant leaves or petals [5,6,7]. 

Proanthocyanidins are flavonoid compounds formed by carbon–carbon bond polymerization with flavane-3-alcohol as the structural unit. Proanthocyanidins can exist as monomers, such as catechin, epicatechin, gallocatechin, and epigallocatechin, or in monomers, dimers, and polymers forms. Except that the condensation mechanism of proanthocyanidins has not been clarified, the biosynthetic pathway of proanthocyanidins has been basically clear; it mainly involves the phenylpropanoid metabolism and flavonoid metabolism. Their synthesis is first regulated by phenylalanine ammonia lyase (PAL) and uses primary metabolic energy to catalyze the phenylpropane metabolism and phenylpropane metabolism [8]. Then, the basic skeleton of flavonoids is formed in chalcone isomerase (CHI). Finally, colorless anthocyanins are formed under the action of dihydroflavonol 4-reductase (DFR) [9], and proanthocyanidins monomers (flavan-3-ol and (+)-catechin) are synthesized by leucoanthocyanidin reductase (LAR) [10]. Under the condition of enhanced UV-B radiation, the increase in the expression of genes encoding chalcone synthase, flavanone 3-hydroxylase, DFR, and anthocyanin synthase in the proanthocyanidins anabolic pathway can result in the increase in the proanthocyanidins content of bilberry leaves [11]. UV-B radiation also promotes the synthesis and accumulation of proanthocyanidins in the peel of agricultural products, such as the UV-B radiation treatment of picked peaches. After 36 h of treatment, the contents of flavane-3-ol and proanthocyanidins in peach peel increased nearly twice [12]. There are few studies on the biological functions of proanthocyanidins with different polymerization degrees because it is not easy to determine the bioavailability of these complex molecules [13]. Studies have shown that the monomers (A-type), dimers (B-type), and trimers (C-type) of proanthocyanidins are easily bioabsorbed [14]. At present, based on the antioxidant, anti-inflammatory, and other health effects of proanthocyanidins [15], attention has been paid to the response of the proanthocyanidins metabolism and antioxidant capacity in crop to UV-B radiation.

Rice is one of the most important food crops in the world, and the most common type (>85%) is white hulled. There are also colored hulls, such as red, black, and purple, which is a kind of healthy food [16]. At present, about 10% of China’s rice-planting resources belong to colored rice, including 8963 red rice germplasm, ranking first in colored rice [17]. Yuanyang terrace is a high-altitude and sustainable paddy agricultural ecosystem [18]. ‘Baijiaolaojing’ is a traditional colored rice planted in Yuanyang terrace. The main color substance of red rice is proanthocyanidins [19]. Proanthocyanidins are water-soluble, with non-toxic and non-allergic characteristics, and are natural free radical scavengers and antioxidants. Many nutrition and drug research results show that proanthocyanidins and their monomers (flavan-3-ol, such as catechins and epicatechin) are important substances that have multiple benefits for human health, belong to antioxidants, and can effectively eliminate free radicals. Proanthocyanidins A2, B2, and C1 were found in red rice and black rice [20]. The intake of proanthocyanidins from food can effectively eliminate oxygen free radicals in the human body [21]. It also has the functions of immune regulation, anti-inflammation, anti-cancer, anti-allergy, anti-microorganism, cardiovascular protection, and anti-thrombosis [22,23].

Aging is the final stage of death in the life cycle of all organisms. The mechanism of aging and its physiological and biochemical changes are very complex. According to the free radical theory, superoxide and other free radicals will damage the cell [24]. Oxygen free radicals, such as superoxide anion and hydroxyl radical, have obvious destructive effects on plasma membranes and cells, causing damage to biological structures, such as lipid membrane degeneration, chromosome heterotopia, molecular breakage, tissue destruction, etc., and finally causing the aging of the body [25]. The hippocampus of brain tissue is an area directly involved in learning and memory. Area CA1 is the main organ that plays the function of the hippocampus of the brain. It is more susceptible to the harmful effects of aging and more sensitive to oxidative stress and apoptosis [26]. An important performance in the aging process is brain aging, which then evolves into chronic inflammation of the nervous system. This is mainly related to the increase in free radicals in brain tissue. The steady state of the microenvironment of neurons is broken by free radicals, resulting in internal environment disorder and the degeneration, necrosis, and apoptosis of neurons [27]. Therefore, the level of oxidative stress, such as the malondialdehyde content and antioxidant enzyme activity, has also become an important indicator of anti-aging [28]. Malondialdehyde (MDA) is a kind of lipid peroxidation metabolite in the body. Generally, MDA is used to reflect the lipid peroxidation metabolism in the body. Superoxide dismutase (SOD) and catalase (CAT) can effectively remove the active oxygen groups in the body, protect the cell membrane system, and is an important member of the organism’s antioxidant system, which can significantly reduce the content of MDA [29]. Reducing the level of oxidative stress and maintaining the redox balance contributes to health and longevity. A high dose of D-galactose can induce oxidative stress in vivo, simulating the natural aging process of mice. The long-term injection of D-galactose into rodents has also become an animal model for antioxidant research [30].

The antioxidant potential of proanthocyanidins is 20 times greater than vitamin E and 50 times greater than vitamin C. It is an effective natural antioxidant to eliminate free radicals in organisms [31]. The phenolic hydroxyl group contained in proanthocyanidins is the main active factor of antioxidation [32]. At present, the research on red rice proanthocyanidins mainly focuses on the degree of polymerization components [19], antioxidant physiological activities [33], genetic analysis [34], and synthetic regulation [35], and there are few studies on environmental adaptation and biological function. In addition, most studies focus on the content of proanthocyanidins and the antioxidant function in blueberries [36] and grapes [37], and there are few studies on proanthocyanidins in crops. In order to understand the effect of UV-B radiation on the synthesis of proanthocyanidins and the antioxidant capacity in red rice, we conducted field experiments by artificially simulating the enhancement of UV-B radiation (0, 2.5, 5.0 and 7.5 kJ·m^−2^·d^−1^) and analyzed the response of the morphology, proanthocyanidins content, and proanthocyanidins synthesis of red rice to the enhancement of UV-B radiation. The effects of UV-B radiation on the antioxidant capacity of rice were evaluated by feeding aging model mice. We hypothesized that UV-B radiation induced proanthocyanidins synthesis in red rice to improve its antioxidant capacity.

## 2. Results

### 2.1. Effect of Enhanced UV-B Radiation on Grain Morphology of Red Rice

UV-B radiation had a significant effect on the grain morphology of red rice (Table 1). With the increase in the UV-B radiation intensity, the 1000 seeds weight, grain length, grain width, and empty grain rate of ‘baijiaolaojing’ decreased in varying degrees. When the UV-B radiation increased to 2.5 kJ·m^−2^·d^−1^, the 1000 grain weight decreased significantly. With the increase in the UV-B radiation intensity, the inhibitory effect was more obvious. When the UV-B radiation intensity increased by 7.5 kJ·m^−2^·d^−1^, the 1000-seeds dry weight and fresh weight decreased by 7.03% and 9.01%, respectively. Under the influence of UV-B radiation, the empty rate of rice grain decreased.

The effect of enhanced UV-B radiation on the maximum natural cross-sectional morphology of rice grain was observed under the magnification of 30 times of a scanning electron microscope (Figure 1). The central area of the rice grain endosperm cells under natural light was slender endosperm cells, the endosperm cells near the edge were thick and short, and there were a small amount of dense starch grains. There were a large number of deep and shallow cracks in the grain, and the seed coat was thick and tightly wrapped around the endosperm cells. Under UV-B radiation treatment, a large number of slender endosperm cells appeared in rice grains, a small number of coarse and short endosperm cells were at the edge, there were a few dense starch grains between the cells; there were no obvious deep cracks on the grains, only a few shallow cracks, the seed coat was thick, and it was widely separated from the endosperm.

Under 500 times magnification of the scanning electron microscope, the grain morphology near the seed coat of rice grain under UV-B radiation treatment was observed. The enhancement of UV-B radiation had a significant effect on the aleurone layer cells and central endosperm starch storage cells of rice grain. Under natural light, there were a large amount of substances in aleurone layer cells and a large number of aleurone grains. The starch granules in the starch storage cells of central endosperm were polyhedral, the arrangement of the starch granules was loose, and there were large gaps. With the enhancement of UV-B radiation, the number of aleurone particles decreased significantly. Only a small amount of aleurone particles existed in aleurone layer cells and there was no obvious gap between the starch particles. The starch particles in central endosperm starch storage cells were closely arranged and there was no gap between the starch particles.

### 2.2. Effect of Enhanced UV-B Radiation on Proanthocyanidin with Different Polymerization Degrees Content in Red Rice

As shown in Figure 2, the content of proanthocyanidin C1 was significantly higher than that of proanthocyanidins A2 and B2. UV-B radiation changed the content of proanthocyanidins B2 and C1 in red rice. Enhanced UV-B radiation significantly increased the content of proanthocyanidin B2 and C1 in rice grains. With the increase in the UV-B radiation intensity, the content of proanthocyanidin B2 and C1 first increased and then decreased. At 5 kJ·m^−2^·d^−1^, the content of proanthocyanidin B2 and C1 in red rice reached the maximum value.

### 2.3. Effect of Enhanced UV-B Radiation on the Activities of Key Enzymes for Proanthocyanidins Synthesis in Red Rice

Enhanced UV-B radiation had a significant effect on the key enzymes of proanthocyanidins synthesis (Figure 3). The 2.5 and 5.0 kJ·m^−2^·d^−1^ treatment significantly inhibited the activities of PAL and CHI. The 7.5 kJ·m^−2^·d^−1^ treatment significantly increased the activity of PAL and had no significant effect on CHI. 2.5 and the 5.0 kJ·m^−2^·d^−1^ treatment had no significant effect on the DFR activity. The 7.5 kJ·m^−2^·d^−1^ treatment significantly inhibited the DFR activity. The 5.0 kJ·m^−2^·d^−1^ treatment significantly increased the LAR activity. The activity of LAR decreased significantly in the 7.5 kJ·m^−2^·d^−1^ treatment.

### 2.4. Morphological Changes in Hippocampus in Mice Brain

It can be seen from Figure 4 that there are 3–4 layers of pyramidal cells in the hippocampus CA1 of mice in the CK group, which are closely arranged with a clear cell morphology and obvious nucleolus. The pyramidal neurons are round and large nuclei with clear nucleolus. Compared with CK, the pyramidal cells in hippocampus CA1 treated with R had 4–5 layers, were arranged more neatly and tightly, with a complete cell structure and morphology, clear nucleolus, increased number of nerve cells, and little apoptosis and karyopyknosis. In the aging model feeding a general food group, pyramidal cells in the hippocampal CA1 area were arranged loosely and disorderly, and the shape of some of the nerve cells changed, showing an irregular dentate shape. Compared with the T group, the pyramidal cells of TR, TRL, and TRM were more orderly arranged, the cell morphology and structure were more complete, and the boundary was clear. Compared with TR, TRL, and TRM, the cells in the TRH group became loose and the number of cells decreased.

### 2.5. Effect of UV-B Radiation on Antioxidant Capacity of Red Rice

Compared with CK, the CAT activity in the serum, liver, brain, and heart of aging model mice decreased significantly and the MDA content increased significantly, indicating that the aging model was successfully constructed (Table 2). The activities of CAT and SOD in the serum, liver, and heart of young mice fed with red rice mouse food were significantly increased. Compared with the aging group, the SOD activity in each organ of mice fed red rice with a different UV-B radiation intensity increased significantly. Compared with T, the CAT and SOD activities of TRM increased significantly, while the MDA content decreased significantly. Compared with male mice, the MDA content in brain and heart of female mice treated with R was significantly higher than that of male mice, while the MDA content in the brain of female mice treated with TRH was significantly lower than that of male mice.

### 2.6. Comprehensive Quantitative Evaluation of Red Rice Antioxidant Capacity

To analyze the changes in the antioxidant capacity of red rice after different UV-B radiation treatments, the antioxidant enzyme activities of the serum, liver, brain, and heart of mice in different feeding schemes were comprehensively and quantitatively evaluated (Table 3). Ranking the comprehensive evaluation values showed that the antioxidant capacity of each treatment was R>TRM>TR>CK>TRL>TRH>T. Red rice has a significant free radical scavenging and antioxidant capacity, and after the 5.0 kJ·m^−2^·d^−1^ treatment, red rice has the best anti-aging effect on the aging model.

## 3. Discussion

Plants use the energy of 400–700 nm wavelengths in solar radiation for photosynthesis, and the light of 280–730 nm wavelengths can induce the construction of plant photomorphogenesis and regulate the process of growth and development. The research on the effects of UV-B radiation on plants has been most in-depth, but the research has not been deeply combined with the crop production system [38].

### 3.1. UV-B Radiation and Red Rice Structure

The morphological structure of red rice grains was changed by UV-B radiation. During the progress of aleurone layer cells in rice grains, large vacuoles were gradually transformed into small protein storage vacuoles, which were filled with protein and minerals and then transformed into aleurone grains [39]. Under enhanced UV-B radiation, the number of aleurone particles in aleurone layer cells decreased significantly, which may lead to the decrease in the nutrient content. Aleurone layer cells have a certain nutrient storage capacity [40] and can store minerals, proteins, and other nutrients [41]. The aleurone layer has the function of nutrient transport. The filling material in the exoplast is transported to the endosperm through the aleurone layer [42]. The enhancement of UV-B radiation inhibited the nutrient transport of plants, resulting in the poor development of the aleurone layer and the inhibition of grain development, which is consistent with the fact that UV-B radiation can inhibit the biomass and growth of rice. However, the enhanced UV-B radiation makes the arrangement of rice starch more compact, reduces the cracks, the rice quality improves, the stress resistance is higher, the gap between the seed coat and endosperm appears, and the seed coat becomes thinner. The decrease in grain cracks indicates that rice starch becomes compact, the relative volume becomes smaller, and cracks appear. In the process of grain aging, proanthocyanidins can slow down the evaporation of water, increase the water holding capacity of plants, and help to maintain the microstructure of starch [43]. In this experiment, under the grain scanning electron microscope, the aging status of starch is significantly different among different treatments with the increase in the UV-B radiation intensity, which may be due to the difference in the proanthocyanidins content in grains between different treatments. Starch aging could significantly affect the physical and cooking characteristics of rice; the oligomeric procyanidins from lotus seedpod had an inhibitory effect on rice starch retrogradation [44]. 

### 3.2. UV-B Radiation and Proanthocyanidin Synthesis in Red Rice

UV-B radiation stress will cause plants to constitute a systematic defense system and synthesize proanthocyanidins that can effectively filter UV-B radiation. During the growth and development of rice, the content of proanthocyanidins in grains was determined by many factors, such as biosynthesis, degradation, the conversion rate, or dilution caused by the accumulation of other compounds. The results showed that the content of proanthocyanidins in rice grains treated with low-intensity (2.5 and 5.0 kJ·m^−2^·d^−1^) UV-B radiation was significantly higher than that in the control group, indicating that rice grains began to accelerate the synthesis of proanthocyanidins B2 and C1 under the induction of UV-B radiation within the tolerance range. Under high-intensity (7.5 kJ·m^−2^·d^−1^) UV-B radiation treatment, the content of proanthocyanidin B2 and C1 decreased significantly, indicating that high-intensity radiation exceeding the tolerance of rice affected the normal physiological activities and inhibited the synthesis of the proanthocyanidins. The research on the response of the soybean flavonoid content to UV-B radiation also found a consistent law. The results show that the change in the flavonoid content depends on the radiation dose. An appropriate dose of UV-B radiation is conducive to the accumulation of soybean flavonoid, while an excessive dose of UV-B radiation may have the opposite effect [45]. The effect of UV-B radiation on the yield of plant secondary metabolites mainly depends on the induction intensity. A high-dose UV-B treatment may cause irreversible damage to the physiological activities due to excessive radiation energy being absorbed by cellular macromolecules such as protein, nucleic acid, and lipid [46]. 

The regulation of light on proanthocyanidins biosynthesis is mainly completed by controlling the phenylalanine metabolism [47]. The increase in phenylalanine products can induce the production of phenolic compounds and anthocyanins through the flavonoid biosynthesis pathway [48]. It was found that the activities of PAL and CHI in a low-intensity UV-B radiation treatment were significantly lower than those in a high-intensity UV-B radiation treatment. This may be because the substrate for PAL and CHI to synthesize proanthocyanidins skeleton under a low intensity meets the needs of the plant’s defense. Sufficient precursors are prepared for proanthocyanidins synthesis at the early stage of grain development. UV-B radiation can induce rice to synthesize more precursors. The same phenomenon is also found in the study of apple maturity [49]. DFR and LAR are the key enzymes for the synthesis of proanthocyanidins. Low-intensity (2.5 and 5.0 kJ·m^−2^·d^−1^) UV-B radiation increased the activities of DFR and LAR in grains, and high-intensity UV-B radiation inhibited the activities of DFR and LAR. The synthesis precursors of proanthocyanidins such as epicatechin are synthesized in the cytoplasm and then transferred to vacuoles through mate encoded by the *TT12* gene. The proton pump was encoded by *AHA10* and glutathione S transferase was encoded by *TT19*. Studies have shown that DFR and LAR, the key enzymes of the proanthocyanidins synthesis substrate, are located in the cytoplasm enzymes related to proanthocyanidins transport and were found in vacuoles [50,51,52]. Rice tissue exposed to low-intensity UV-B radiation can cause reversible and elastic “positive stress” (a temporarily activated stress response, which makes plant cells in a low alert state by stimulating specific signal pathways, to activate the metabolism of the secondary defense system and promote the synthesis of proanthocyanidins). When the level of exposure to UV-B radiation exceeds the tolerance limit, irreversible “stress dilemma” (strong stress conditions under environmental conditions that are seriously unfavorable to growth, resulting in plant metabolic damage) [53], and proanthocyanidins synthesis is blocked. The content of proanthocyanidins B2 and C1 in rice grains was the highest when treated with 5.0 kJ·m^−2^·d^−1^ UV-B radiation, indicating that the radiation intensity can achieve the balance between “forward stress” and “stress dilemma” of the maximum accumulation of rice grains. The effect of UV-B radiation on proanthocyanidins synthesis during rice grain development needs to be further studied.

### 3.3. UV-B Radiation and Antioxidant Capacity of Red Rice

The experiment of constructing aging model mice showed that red rice has a certain application value in the treatment and intervention of aging-related diseases. Brain aging is associated with chronic inflammation of the nervous system. Senile chronic injury breaks the homeostasis of the microenvironment where neurons live, leading to the imbalance of inflammatory cytokines and the related inhibitors, causing the degeneration, necrosis, and apoptosis of neurons [54]. Hippocampus CA1 is the most sensitive to external pathological factors and it is also the place where pathological changes first occur. Compared with the aging model, the morphology and structure of CA1 cells were more complete after adding red rice, and the apoptosis was significantly improved. Proanthocyanidins inhibits the p38 MAPK pathway, suppresses the inflammation in hippocampus, and increase the externalization of AMPAR [55]. 

The oxidative stress reaction is one factor that causes central nervous system, liver, and heart diseases during aging [56]. Reactive oxygen species (ROS) and its intermediates (MDA) are the principal inducements of nerve and organ diseases during aging by oxidizing protein, DNA, and lipid macromolecules. Various endogenous antioxidants (superoxide dismutase, glutathione, glutathione peroxidase, vitamin E, and vitamin C) constitute the human body’s own natural anti-aging system and maintain a certain balance with active oxygen to limit free radical damage to nerves and organs [57]. Oxidative damage during aging may be caused by insufficient antioxidants. Enhanced UV-B radiation can induce rice grains to synthesize proanthocyanidins and the content of proanthocyanidins first increases and then declines with the increase in the radiation intensity, which is consistent with the result of the red rice’s antioxidant capacity. The antioxidant capacity of TRH is lower than TRM, and TRM has strong antioxidant and anti-aging capabilities. This study found that the content of MDA in the aging model mice induced by D-galactose was considerably increased. As a product of lipid peroxidation, the free radicals produced by aging would interact with lipids to produce MDA. Compared with the aging group, the consumption of red rice in aging model mice can reduce the level of MDA in the serum, liver, brain tissue, and heart of mice. This shows that eating red rice can delay aging by reducing the content of MDA in the aging model. Among the relevant mechanisms which proposed regulating aging, the theory linking an aging and cellular oxidative stress reaction is more scientific [58]. Antioxidant-rich diets have been shown to prolong the life span of animal models. The dietary feeding of blueberry extract extended the lifespan of fruit flies and *Caenorhabditis elegans*. Additionally, the expression of SOD and catalase were increased. The beneficial effect of blueberry on prolonging lifespan may be related to the endogenous antioxidant system enhanced by the exogenous antioxidant intake [59]. With the increase in age, the activity of antioxidant enzymes such as SOD and CAT in the body decreases and the ability to scavenge free radicals decreases, leading to a chain reaction of free radicals, which ultimately leads to an increase in malondialdehyde lipid peroxidation end-products, which is consistent with the results of this study [60]. Compared with the control group, the antioxidant enzyme activity of mice fed red rice was significantly increased, and the content of malondialdehyde was significantly reduced, indicating that red rice could improve the antioxidant capacity of mice. UV-B radiation induces plants to secrete chemicals that can effectively filter UV-B radiation, prevent themselves from being injured, improve the antioxidant capacity of plant products, and have broad development prospects in food, medicine, and animal husbandry.

## 4. Materials and Methods

### 4.1. Summary of Test Site

The test site was located in Qingkou village, Xinjie Town, Yuanyang County, Honghe Prefecture, Yunnan Province (23°7′ N, 102°44′ E). The altitude of this area is 1600 m and the background intensity of UV-B radiation is 10 kJ·m^−2^·d^−1^. The local soil is hydragric anthrosols, with a pH value of 6.45, an organic matter content of 26.8 g·kg^−1^, total N of 2.48 g·kg^−1^, total P of 0.74 g·kg^−1^, total K of 6.03 g·kg^−1^, alkali hydrolyzed n of 67.9 m g·kg^−1^, available P of 20.5 m g·kg^−1^, and available K of 150.7 mg·kg^−1^. The cultivated rice variety is ‘baijiaolaojing’ and the seeds were provided by the agricultural science station in Xinjie Town. It has been cultivated locally for more than 300 years and is one of the main cultivated varieties at present.

### 4.2. Field Test Design

Rice planting treatment: ‘baijiaolaojing’ was sown on 20 March 2018 and transplanted to the experimental plot on 15 May. The test site is arranged with a total area of 390 cm × 225 cm, with a 50 cm interval between the plots. Each plot was planted with 10 rows of rice, 10 plants in each row, 30 cm row spacing, and 15 cm plant spacing. Each cluster is a seedling and protective rows are set around. During the growth period of rice, no chemical fertilizers and pesticides are used. The whole growth period is flooded.

UV-B radiation treatment: a lamp rack was installed with an adjustable length above each row of rice and the UV-B lamp tube was erected (40 W, wavelength 280–320 nm, Shanghai Gucun instrument company, Shanghai, China). The radiation intensity was measured at the top of the rice plant with a UV-B radiation meter (photoelectric instrument factory of Beijing Normal University, Beijing, China). Four groups of natural light (0 kJ·m^−2^·d^−1^) and UV-B radiation (2.5, 5.0, 7.5 kJ·m^−2^·d^−1^) were set up, which were equivalent to the local ozone attenuation of 0%, 10%, 20%, and 30%, respectively. The daily irradiation time was 7 h (10:00–17:00) and the UV-B radiation treatment was not carried out on cloudy or rainy days. With the growth of the rice, the height of the lamp holder was adjusted to ensure the radiation intensity was unchanged. On 25 June, the light was turned on at the heading stage of rice and the experiment was stopped after the rice was harvested on 28 September. 

### 4.3. Observation on Morphological Structure of Rice Grain

The rice grains under each treatment were cut to be 0.6 mm thick and washed with 0.1 mol·L^−1^ phosphoric acid buffer with pH = 7.4 for 3 times. The grain was fixed in 2% glutaraldehyde fixed solution (prepared with 0.1 mol·L^−1^, pH = 7.2 sodium phosphate buffer) for 2 h. The fixed seeds were rinsed three times in pH = 7.4 0.1 mol·L^−1^ phosphoric acid buffer for 10–15 min each time. In total, 1% osmic acid was added to the rinsed sample and placed in the refrigerator at 4 °C for 1–2 h. The fixed seeds were rinsed again with pH = 7.4 0.1 mol·L^−1^ phosphoric acid buffer 3 times. Ethanol with concentrations of 30%, 50%, 70%, 80%, 90%, and 100% was added at room temperature for step-by-step dehydration. We dehydrated 3 times with 100% ethanol and the dehydration process time of each stage was 10 min. Isoamyl acetate was added to the dehydrated grain and it was left to stand for 15 min. This step was repeated twice. After the gradual dehydration and critical point drying by the critical point dryer (K850, Quorum Technologies Ltd., Ringmer, UK), the sample was placed close to the double-sided adhesive tape of a conductive carbon film on the sample table of an ion sputtering instrument for gold spraying for about 30 s. A scanning electron microscope (SU8100, Hitachi, Tokyo, Japan) was used to observe, collect, and analyze the images.

### 4.4. Determination of Proanthocyanidin with Different Polymerization Degrees Content

Ultra-high performance liquid chromatography coupled to triple quadrupole mass spectrometry (UPLC-QqQ-MS) was used to determine the content of proanthocyanidins with different polymerization degrees. After grinding rice grains with liquid nitrogen, 0.5 g of the milled sample (accurate to 0.0001) was weighed and placed in a 10.0 mL centrifuge tube. A total of 3.00 mL of 80% methanol water was added. The vortex oscillator (Vortex-5, Kylin-Bell, Xiangyi, China) was used to mix with vortex for 1 min, it underwent an imposed ultrasonic vibration for 40 min, it stood for 60 min, it was mixed by vortex for 30 s, and was then filtered by a 0.22 μm membrane to detect proanthocyanidin A2 and proanthocyanidin B2. Because the content of proanthocyanidin C1 exceeded the detection range, it needed to be diluted 50 times with 80% methanol water before determining.

The preparation of the standard curve: the standard samples of proanthocyanidin A2, proanthocyanidin B2, and proanthocyanidins C1 were purchased from Shanghai Yuanye Biotechnology Co., Ltd. A total of 20.0 µL of standard solution (100 µg·mL^−1^) was accurately taken. It was diluted with 80% methanol water to prepare a standard series solution with a concentration of 2000, 1000, 500, 200, 100, 50.0, 20.0, 10.0, 5.00, 2.00, 1.00, 0.50, 0.20, and 0.10 ng·mL^−1^. The standard curve was drawn according to the peak area of the standard under different concentrations and corresponding concentrations. 

A liquid chromatograph (Waters, St Quentin en Yvelines, France) and a mass spectrometer AB SCIEX 5500 QQQ MS (Toronto, ON, Canada) were used in series, in which the chromatographic column model was Acquire UPLC BEH C18 (1.7 µm, 2.1 mm × 100 mm), the column temperature was 40 °C, and the flow rate was 0.30 mL·min^−1^. The following mobile phase was used: A: water containing 0.1% formic acid; B: 100% acetonitrile. In total, 4 µL sample volumes were taken for gradient elution for 8 min. The sample gradient elution procedure is shown in Table 4.

The optimum operating parameters were determined by the electro spray ionization (ESI) interface in the positive ion mode. The generic mass spectrometry parameters of the analyte were developed and used for the analysis. These parameters were: curtain gas: 20 arb; collision gas: 9 arb; ion spray voltage: −4500 V; ion source temperature: 450 °C; ion source gas 1: 55 arb; ion source gas 2: 55 arb; and the multiple reaction monitoring (MRM) acquisition parameters were according to the above and the chromatographic and mass spectrum conditions in Table 4. The prepared standard solution was added into the sample injection bottle for sample injection. The retention time determined by the substance peak was shown in Table 5. Multi Quant software was used to integrate the peak area of the results and brought it into the standard curve equation to calculate the content.

### 4.5. Determination of Proanthocyanidins Synthase Activity

The preparation of crude enzyme solution: 0.300 g rice grains were ground with 3 mL of 0.1 mol·L^−1^ borate buffer (pH = 8.8, containing 5 mmol·L^−1^ β-mercaptoethanol and 0.1 g·L^−1^ PVP) in an ice water bath and centrifuged at 10,000× *g* for 10 min in a 4 °C refrigerated centrifuge to prepare the crude enzyme solution.

The determination of the phenylalanine ammonia lyase (PAL) activity [61]: 20 μL crude enzyme solution was added with 780 μL of 0.1 mol·L^−1^ boric acid buffer and 200 μL of 0.02 mol·L^−1^ phenylalanine. The reaction solution was put in a constant temperature water bath at 30 °C for 0.5 h and measured the change in the OD290 value with an ultraviolet visible spectrophotometer (UV-5800, Shanghai Yuan Analytical Instrument Co., Ltd., Shanghai, China). Each milligram of tissue protein changes the 290 nm absorbance value to 0.1 unit per minute in each milliliter of the reaction system, which meant one activity unit.

The determination of the chalcone isomerase (CHI) activity [49]: naringin was used as the raw material, acidified after being treated with 50% potassium hydroxide and then ethanol aqueous solution was added to recrystallize to synthesize chalcone. A total of 0.75 mL of crude enzyme solution was diluted to 2 mL by 50 mmol Tris-HCl (containing 50 mM Tris HCl, 50 mM KCN, pH = 7.4). In total, 50 μL of 1 mg·mL^−1^ chalcone was added and bathed in 34 °C water for 30 min. The same amount of crude enzyme solution was boiled in water for 10 min as the control. The change in the absorbance value at the 381 nm wavelength was measured to express the enzyme activity of CHI. One activity unit represents the amount of enzyme required to change the absorbance by 0.001 per minute.

The determination of the dihydroflavonol 4-reductase (DFR) activity: 0.1 mol·L^−1^ Tris-HCl buffer (pH = 7.4), 1 μmol of dihydroquercetin, and 1 μmol of NADPH were mixed with 0.3 mL crude enzyme solution. Dihydroquercetin was catalyzed by the DFR enzyme as the substrate. It was placed in a water bath pot at 30 °C for 1 h. In total, 1 mL of ethyl acetate was added and evaporated with a rotary evaporator at 40 °C. Ethyl acetate was added and evaporated repeatedly. Additionally, then the extract was combined. The residue was washed with distilled water (three times with a total of 0.6 mL of distilled water) to remove ethyl acetate. Colorless anthocyanins can be catalyzed to produce anthocyanins in ethyl acetate extract. A total of 1 mL of butanol hydrochloric acid solvent with a volume ratio of 95:5 was added to the extract under a water bath at 95 °C for 30 min. The absorbance was measured at 550 nm with a UV–VIS spectrophotometer. The DFR enzyme activity was expressed as the amount of enzyme required to decompose dihydroquercetin to produce 1 mmol of catechin per milligram of protein per minute at 30 °C, pH = 7.5.

The determination of the leucoanthocyanidin reductase (LAR) activity: 0.5 mL of 0.1 mol·L^−1^ Tris HCl buffer, respectively (pH = 7.5), 1 µmol·L^−1^ dihydroquercetin, and 1 mmol·L^−1^ NADPH were added to 0.5 mL of crude enzyme under a water bath at 37 °C for 1 h. Then, 2 mL of 8% hydrochloric acid methanol solution and 2 mL of vanillin methanol solution were added under a water bath at 37 °C for 1 h. The absorbance at 500 nm was measured with a UV–VIS spectrophotometer. The LAR enzyme activity was expressed as the amount of enzyme required to decompose dihydroquercetin to produce 1 mmol of catechin per milligram of protein per minute.

### 4.6. Animals Experimental Protocol

Eighty-four SPF Kunming mice aged 4–6 weeks were purchased from Hunan Slake Jingda Experimental Animal Co., Ltd. (Changsha, China, Certificate: SCXK 2019-0004). Half male and half female mice (male weight 30.45 ± 0.45 g, female weight 20.12 ± 0.32 g) were bred in the animal facility of Yunnan Agricultural University. Rice grains were crushed into powder with a grinder when the rice was mature. It was mixed with crushed basic feed in a ratio of 1:1 to ensure the daily nutrition intake of mice The basic feed was provided by the Department of Experimental Zoology of Kunming Medical University, which is mainly composed of wheat bran, soybean flour, and corn flour. The mixed mice diet of red rice treated with different intensities of UV-B radiation was prepared. After an acclimatizing period of 5 days, the mice were randomly divided into seven groups. The mice were divided into the basic feed group (CK), red rice mice food group (R), aging model basic feed group (T), aging model red rice mice food group (TR), aging model 2.5 kJ·m^−2^·d^−1^-treated rice mice food group (TRL), aging model 5.0 kJ·m^−2^·d^−1^-treated rice mice food group (TRM), and aging model 7.5 kJ·m^−2^·d^−1^-treated rice mice food group (TRH). Every 6 mice of the same sex lived in a M3 mouse cage made of polypropylene (318 mm × 202 mm × 135 mm); the top of the cage was equipped with stainless steel wire mesh, the feeding temperature was controlled at 26 ± 1 °C, there was a stable humidity of 45–55%, and day and night alternated for 12 h to ensure an adequate diet and free diet. The mice of T, TR, TRL, TRM, and TRH were subcutaneously injected with 1.25 g·kg^−1^ of D-galactose (Solebar Technology Co., Ltd., Beijing, China) at the back of the neck every afternoon to build an aging model [62]. The mice of CK and R were injected with physiological saline of the same volume for 8 weeks. After the treatment, all the mice were fasted for 12 h and euthanized with CO_2_. After the mice were killed, the thoracic cavity was opened and the whole blood was taken from the abdominal aorta. The blood sample was placed at 4000× *g* for 10 min at 4 °C to obtain the serum. After dissection, the brain of the mouse was taken out and fixed in 4% paraformaldehyde for more than 24 h for dehydration for the anatomical morphology observation. The liver, brain, and heart were quickly frozen in liquid nitrogen within 10 min after death and then stored in a refrigerator at 80 °C. The animal protocol used in this study was reviewed and approved by the Institutional Animal Care and Use Committee of Yunnan Agricultural University with respect to the ethical issues and scientific care.

### 4.7. Brain Tissue Section

At 4 °C, the left brain was fixed in 4% paraformaldehyde for more than 24 h. Coronal brain sections were made using a sliding microtome (5 μm). Then, hematoxylin eosin staining was added to the samples and then rinsed in distilled water. Stepwise, they were dehydrated in graded alcohol (70%, 85%, 95%, and 100%) each for 30 s. Finally, the samples were cleared in xylene. The coronal plane of the brain was scanned with a microscope slide scanner (Panoramic DESK, P-MIDI, P250, 3DHISTECH Ltd., Budapest, Hungary) to observe the morphological characteristics of the hippocampus CA1 region shown by hematoxylin eosin staining.

### 4.8. Determination of Antioxidant Index

The mice liver, brain, and heart were homogenized into 10% tissue homogenate with 4 °C normal saline. The prepared homogenate was centrifuged at 4 °C at 3000 r/min for 15 min. The supernatant and serum were taken to determine the antioxidant index. The CAT, SOD activity, and MDA content were measured using kits from the Nanjing Jiancheng Bioengineering Institute (Nanjing, China).

### 4.9. Comprehensive Quantitative Evaluation of Antioxidant Capacity

The activity indexes of antioxidant enzymes are normalized in a dimensionless way. The dimensionless treatment formula for the CAT and SOD enzyme activities is as follows:(1)$$y_{\mathrm{CAT}/\mathrm{SOD}}=\frac{x_{i}-min}{max-min}$$
where $x_{i}$ is the value of the ith indicator, min is the minimum value of the data in the indicator, and max is the maximum value of the data in the indicator. As MDA is a reverse index, the antioxidant capacity decreases with the increase in the value. Therefore, the dimensionless treatment formula for calculating the MDA content is opposite to the above formula, which is:(2)$$y_{\mathrm{MDA}}=\frac{max-x_{i}}{max-min}$$

Therefore, the dimensionless comprehensive evaluation of the antioxidant indexes of each treatment was carried out, and the comprehensive evaluation value (CEV) of the antioxidant capacity was determined by the average value.

### 4.10. Data Statistical Analysis

The data were sorted by Microsoft Excel and the mean and variance were calculated. The statistical software IBM SPSS Statistics 20.0 (SPSS Inc., Chicago, IL, USA) was used to analyze the differences between the treatments by single factor difference analysis and Duncan test (*p* < 0.05). Draw with origin 9.1.

## 5. Conclusions

UV-B radiation induces the synthesis of proanthocyanidins B2 and C1 in red rice. The neuronal damage in the hippocampus of aging model mice was improved by red rice. The antioxidant capacity of red rice was significantly improved by proanthocyanidins synthesis induced by 5.0 kJ·m^−2^·d^−1^ UV-B radiation.

## Figures and Tables

**Figure 1 ijms-24-03397-f001:**
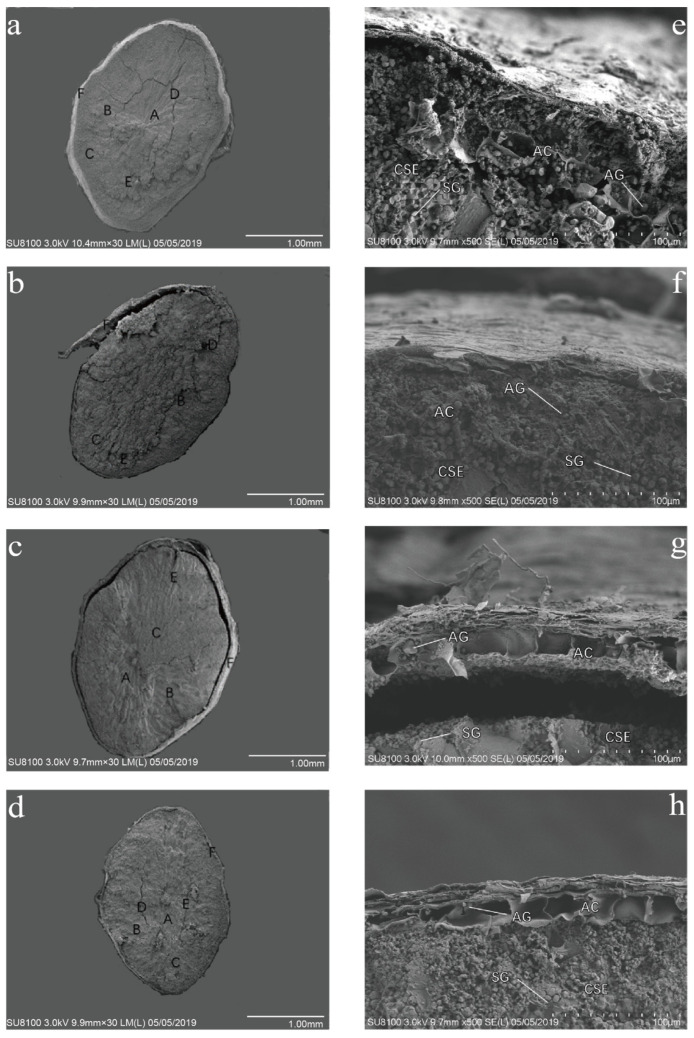
Changes in grain anatomical structure of rice under UV-B radiation treatment: 30 times of microscope magnification (**a**–**d**), 500 times of microscope magnification (**e**–**h**), natural light (**a**,**e**), 2.5 kJ·m^−2^·d^−1^ (**b**,**f**), 5.0 kJ·m^−2^·d^−1^ (**c**,**g**), 7.5 kJ·m^−2^·d^−1^ (**d**,**h**). A: slender endosperm cells. B: thick and short endosperm cells. C: dense starch grains. D: deep cracks. E: shallow cracks. F: seed coat. AC: aleurone layer cells. AG: aleurone granules. CSE: central endosperm starch storage cells. SG: starch granules.

**Figure 2 ijms-24-03397-f002:**
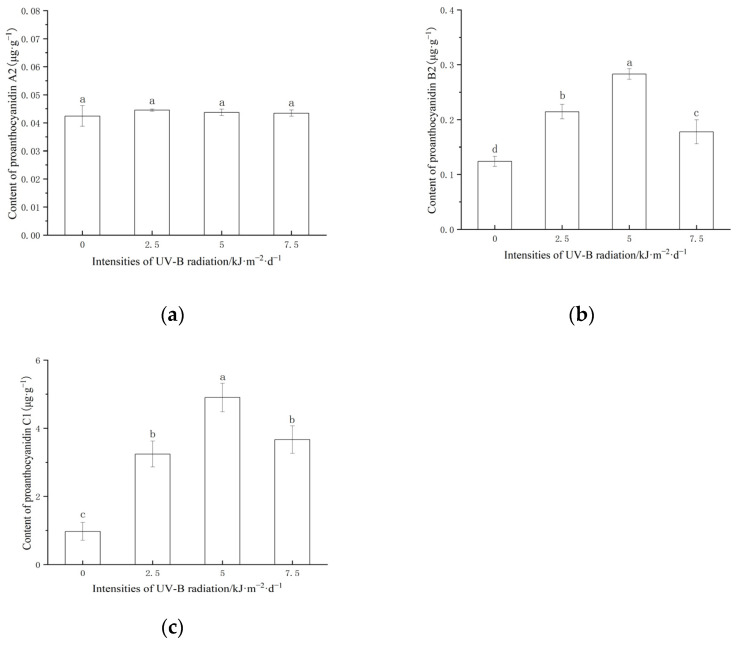
Content of proanthocyanidins A2 (**a**), B2 (**b**), and C1 (**c**) in red rice under different UV-B radiation. Data are mean ± SE (n = 3). The different lowercase letters indicate significant differences among treatments (*p* < 0.05).

**Figure 3 ijms-24-03397-f003:**
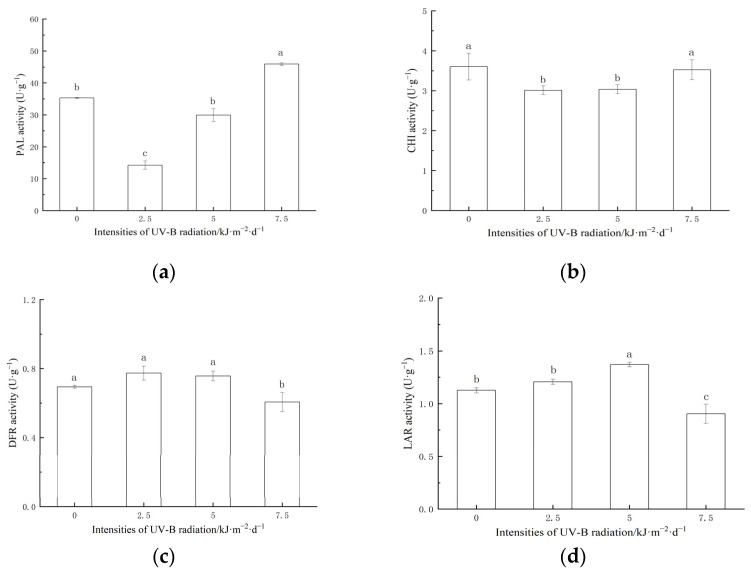
Effects of enhanced UV-B radiation on PAL (**a**), CHI (**b**), DFR (**c**), LAR (**d**) activities in red rice. Data are mean ± SE (n = 3). The different lowercase letters indicate significant differences among treatments (*p* < 0.05).

**Figure 4 ijms-24-03397-f004:**
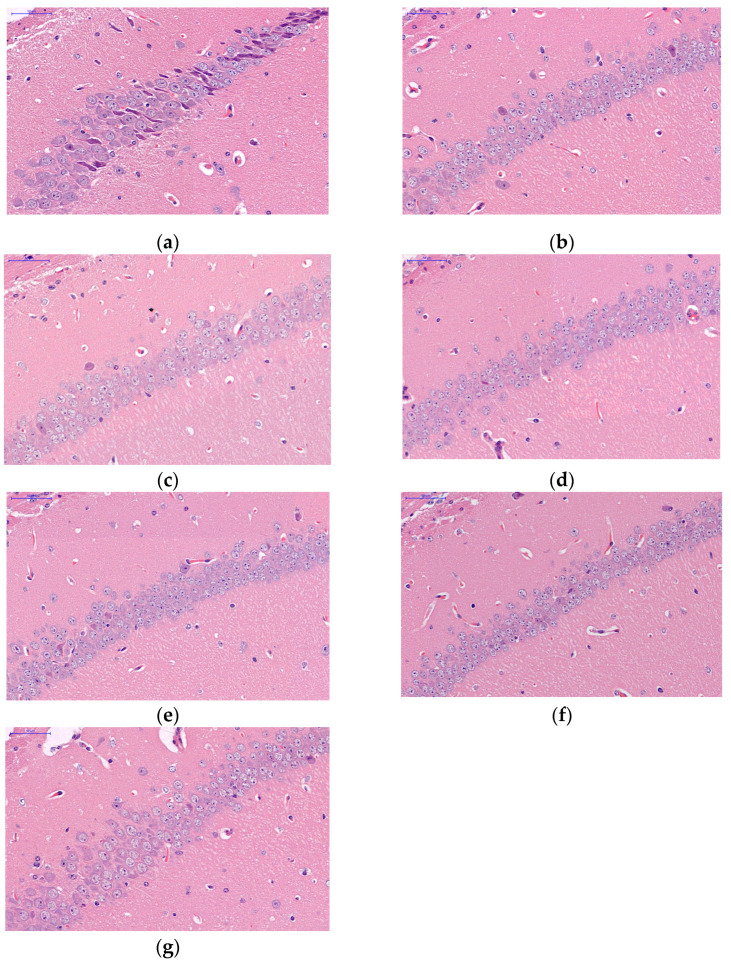
Effects of feeding red rice treated by UV-B radiation on hippocampus CA1 of mice brain. (**a**) CK: the basic feed group, (**b**) R: red rice mice food group, (**c**) T: aging model basic feed group, (**d**) TR: aging model red rice mice food group, (**e**) TRL: aging model 2.5 kJ·m^−2^·d^−1^-treated red rice mice food group, (**f**) TRM: aging model 5.0 kJ·m^−2^·d^−1^-treated red rice mice food group, (**g**) TRH: aging model 7.5 kJ·m^−2^·d^−1^-treated red rice mice food group.

**Table 1 ijms-24-03397-t001:** Effects of enhanced UV-B radiation on grain morphology of ‘baijiaolaojing’.

Treatment (kJ·m^−2^·d^−1^)	1000-Seeds Dry Weight (g)	1000-Seeds Fresh Weight (g)	Moisture Content (%)	Grain Length/mm	Grain Width/mm	Empty Grain Rate (%)
0	22.05 ± 0.51 a	27.75 ± 1.05 a	25.85 ± 3.64 ab	8.05 ± 0.55 a	3.54 ± 0.35 a	17.01 ± 2.24 a
2.5	21.35 ± 0.56 ab	28.50 ± 0.30 a	33.53 ± 2.43 a	7.21 ± 0.46 b	3.47 ± 0.30 ab	10.42 ± 1.43 b
5.0	20.81 ± 0.72 b	27.50 ± 1.00 a	32.34 ± 7.43 ab	7.21 ± 0.41 b	3.32 ± 0.39 ab	14.53 ± 1.47 ab
7.5	20.50 ± 0.36 b	25.25 ± 0.55 b	23.22 ± 4.84 b	7.19 ± 0.68 b	3.27 ± 2.80 b	10.69 ± 1.95 b

Data are mean ± SE (n = 10). Different letters within a column indicate significant differences among different treatments (*p* < 0.05, n = 10).

**Table 2 ijms-24-03397-t002:** Effects of feeding red rice treated by UV-B radiation on antioxidant indexes of mice organs.

Treatment	Gender	CAT(U·g^−1^)				SOD(U·g^−1^)				MDA(μmol·g^−1^)			
		Serum	Liver	Brain	Heart	Serum	Liver	Brain	Heart	Serum	Liver	Brain	Heart
CK	Male	25.65 ± 3.86 ef	17.05 ± 4.52 cd	82.38 ± 4.81 abc	45.73 ± 5.76 cd	272.77 ± 3.95 b	217.51 ± 3.39 d	180.46 ± 6.41 e	279.93 ± 12.49 de	5.64 ± 1.31 cde	12.23 ± 1.55 e	14.64 ± 0.78 ef	25.87 ± 1.82 b
	Female	28.84 ± 3.86 de	19.92 ± 1.46 c	83.01 ± 1.46 abc	42.86 ± 8.67 cd	278.76 ± 11.77 b	227.67 ± 4.80 d	186.28 ± 5.55 e	314.56 ± 21.17 c	7.22 ± 1.57 bc	14.92 ± 1.23 cde	12.7 ± 0.80 f	16.93 ± 3.26 efg
R	Male	37.44 ± 2.41 bc	31.71 ± 2.92 a	90.98 ± 6.78 a	94.48 ± 7.96 a	299.78 ± 1.94 a	259.95 ± 8.30 c	254.25 ± 11.89 b	376.55 ± 11.07 b	4.99 ± 0.12 def	15.31 ± 3.20 cde	18.26 ± 1.24 d	13.63 ± 2.41 g
	Female	35.85 ± 2.87 bc	25.33 ± 0.96 b	85.56 ± 8.60 ab	91.30 ± 1.66 a	316.28 ± 11.65 a	264.90 ± 7.30 bc	268.57 ± 2.20 b	256.47 ± 5.43 ef	4.82 ± 0.16 defg	14.61 ± 1.15 cde	21.57 ± 1.43 c	17.87 ± 1.56 def
T	Male	18.97 ± 3.07 g	12.91 ± 2.53 efg	65.17 ± 3.36 de	31.71 ± 5.76 e	179.81 ± 3.87 g	124.63 ± 9.59 e	101.72 ± 4.36 f	148.53 ± 16.19 h	13.51 ± 1.11 a	27.47 ± 9.96 a	30.53 ± 1.20 a	28.54 ± 0.95 a
	Female	21.83 ± 4.72 fg	10.04 ± 0.96 gh	53.06 ± 7.65 e	19.28 ± 4.91 f	186.66 ± 5.52 g	123.98 ± 3.86 e	101.75 ± 6.99 f	153.24 ± 9.97 h	14.53 ± 0.62 a	23.99 ± 2.50 ab	31.12 ± 3.09 a	29.97 ± 1.68 a
TR	Male	49.87 ± 5.44 a	14.50 ± 1.46 def	92.89 ± 8.35 a	45.09 ± 6.14 cd	260.81 ± 5.01 bcd	256.90 ± 9.78 c	215.14 ± 7.12 cd	273.49 ± 7.55 de	7.60 ± 2.14 b	13.22 ± 1.30 de	25.68 ± 2.73 b	24.18 ± 0.86 bc
	Female	46.68 ± 2.41 a	14.18 ± 1.46 def	90.98 ± 9.67 a	47.64 ± 2.76 cd	253.35 ± 7.47 cde	254.41 ± 4.48 c	195.11 ± 8.71 de	272.99 ± 10.78 de	6.08 ± 1.17 bcd	13.36 ± 1.36 de	25.67 ± 1.04 b	14.92 ± 3.11 gh
TRL	Male	37.12 ± 1.99 bc	10.68 ± 0.55 fgh	54.65 ± 2.41 e	51.46 ± 6.71 c	270.90 ± 9.97 bc	257.41 ± 5.06 c	180.73 ± 4.90 e	290.60 ± 10.03 d	3.00 ± 1.07 g	20.08 ± 1.39 bc	21.47 ± 0.95 c	22.59 ± 1.42 bc
	Female	39.67 ± 2.53 b	13.22 ± 1.46 efg	55.93 ± 4.17 e	44.45 ± 4.17 cd	308.46 ± 2.57 a	321.61 ± 12.78 a	188.41 ± 2.24 e	284.69 ± 12.66 d	3.95 ± 1.04 efg	18.14 ± 2.74 bcde	21.76 ± 0.55 c	17.46 ± 0.70 efg
TRM	Male	36.81 ± 2.53 bc	12.27 ± 3.07 efg	74.09 ± 8.33 bcd	79.82 ± 5.74 b	251.89 ± 8.91 de	266.37 ± 12.74 bc	228.24 ± 12.89 c	372.26 ± 16.75 b	3.56 ± 0.48 fg	16.49 ± 1.89 cde	23.97 ± 0.77 bc	21.67 ± 2.26 cd
	Female	33.30 ± 2.41 cd	15.14 ± 1.46 de	82.38 ± 4.91 abc	76.64 ± 7.73 b	236.03 ± 6.95 e	276.78 ± 18.86 b	331.37 ± 36.62 a	432.74 ± 25.96 a	3.88 ± 0.23 efg	19.61 ± 1.84 bcd	17.10 ± 1.79 de	20.68 ± 2.65 cde
TRH	Male	25.02 ± 4.31 ef	7.17 ± 0.96 h	64.21 ± 3.86 de	38.40 ± 7.67 de	237.66 ± 2.21 e	220.67 ± 4.98 d	183.97 ± 7.50 e	233.30 ± 14.02 f	3.01 ± 0.24 g	23.74 ± 4.31 ab	14.75 ± 0.19 ef	22.03 ± 1.89 bc
	Female	26.93 ± 1.99 ef	11.95 ± 0.96 efg	71.86 ± 10.53 cd	38.72 ± 2.87 de	211.01 ± 28.23 f	233.14 ± 5.00 d	192.20 ± 4.13 e	201.86 ± 9.36 g	3.74 ± 0.66 efg	17.97 ± 1.79 bcde	13.51 ± 0.33 f	13.59 ± 3.85 g

CK: the basic feed group, R: red rice mice food group, T: aging model basic feed group, TR: aging model red rice mice food group, TRL: aging model 2.5 kJ·m^−2^·d^−1^-treated rice mice food group, TRM: aging model 5.0 kJ·m^−2^·d^−1^-treated rice mice food group, TRH: aging model 7.5 kJ·m^−2^·d^−1^-treated rice mice food group. Data are mean ± SE (n = 3). Different letters within a column indicate significant differences among different treatments (*p* < 0.05, n = 3).

**Table 3 ijms-24-03397-t003:** Comprehensive quantitative evaluation of antioxidant capacity of red rice under UV-B radiation.

Treatment	CAT	SOD	MDA	CEV	Rank
CK	0.45	0.52	0.77	0.58	4
R	0.83	0.73	0.78	0.78	1
T	0.11	0.01	0.05	0.06	7
TR	0.64	0.53	0.63	0.60	3
TRL	0.32	0.62	0.66	0.53	5
TRM	0.55	0.72	0.67	0.65	2
TRH	0.24	0.37	0.77	0.46	6

CK: the basic feed group, R: red rice mice food group, T: aging model basic feed group, TR: aging model red rice mice food group, TRL: aging model 2.5 kJ·m^−2^·d^−1^-treated rice mice food group, TRM: aging model 5.0 kJ·m^−2^·d^−1^-treated rice mice food group, TRH: aging model 7.5 kJ·m^−2^·d^−1^-treated rice mice food group, CEV: comprehensive evaluation value.

**Table 4 ijms-24-03397-t004:** Sample gradient elution procedure.

Time (min)	Flow Rate (mL·min^−1^)	Acetonitrile Solvent Gradient (%)	Curve
Initial	0.30	10	Initial
2.00	0.30	10	6
6.00	0.30	90	6
7.00	0.30	90	6
7.10	0.30	10	6
8.00	0.30	10	6

**Table 5 ijms-24-03397-t005:** Mass spectrum MRM acquisition parameters.

Compounds	Q1 Mass (Da)	Q3 Mass (Da)	Retention Time (min)	Declustering Potential (V)	Collision Energy (V)	Cell Exit Potential (V)
Proanthocyanidin A2	575.1	285	4.03	−190	−38	−9
Proanthocyanidin A2	575.1	448.9	4.03	−190	−30	−14
Proanthocyanidin A2	575.1	539.1	4.03	−199	−35	−11
Proanthocyanidin B2	577	289	3.30	−171	−36	−8
Proanthocyanidin B2	577	407.1	3.30	−172	−33	−8
Proanthocyanidin B2	577	425.1	3.30	−173	−20	−13
Proanthocyanidin C1	865.1	287	3.67	−48	−43	−8
Proanthocyanidin C1	865.1	425	3.67	−25	−39	−13
Proanthocyanidin C1	865.1	695	3.67	−13	−35	−14

## Data Availability

Not applicable.

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
