# Peer review of "Enhanced UV-B Radiation Induced the Proanthocyanidins Accumulation in Red Rice Grain of Traditional Rice Cultivars and Increased Antioxidant Capacity in Aging Mice"

_ijms, 2023, doi:10.3390/ijms24043397_

Round 1

Reviewer 1 Report

The manuscript entitled "Enhanced UV-B radiation induced the proanthocyanidins accumulation in red rice grain of traditional rice cultivars and increased antioxidant capacity in aging mice" described about the potential increase in the anthocyanin contents of traditional rice cultivars exposed to different UV-B irradiation. The manuscript looks well-structured, however, require some modifications. Please see the comments as given below;

1. There is no information on the background of the study in the abstract section. It needs to be included

2. There is not much information about the animal model studies in the abstract

3. In the introduction, authors needs to provide an explanation for the question "Why UV-B compared to other irradiations?". A brief description is highly recommended.

4. Why the authors emphasized the proanthocyanidins alone? What about the concentrations of other phenolic/ flavonoid compounds? Authors needs to indicate a satisfactory explanation for choosing one class of compound alone. Total polyphenol and flavonoid contents needs to be included.

5. Why the data in table 3 was not represented as mean and standard deviation

6. All the tables and figures needs to be self-explanatory. If authors included any non-standard abbreviations, it must be explained in the respective legends. Likewise, the statistical notations like a,b, *,# etc must be mentioned clearly in the legends

7. What does the table 2 title means? Authors needs to be more careful before submission

8. Authors analyzed the enzymatic antioxdants in mice; why the non-enzymatic antioxidants like glutathione etc was not evaluated. Also, the histological analysis might have provided more information.

9. Authors could have analyzed the composition of the total isolated proanthocyanidins. I recommend a LCMS analysis for this

10. The conclusion needs to be expanded more.

Reviewer 2 Report

In the manuscript Enhanced UV-B radiation induced the proanthocyanidins accumulation in red rice grain of traditional rice cultivars and increased antioxidant capacity in aging mice, authors Xiang Li, Jianjun Sheng, Zuran Li, Yongmei He, Yanqun Zu and Yuan Li,  hypothesized that UV-B radiation induced proanthocyanidins synthesis in red rice to improve its antioxidant capacity.

Abstract

L 13 …filed(0, 2.5, 5.0, 7.5 kJ·m-2·d-1). Please, write space after field and units as they should be!

L 13-14 The following sentence is not clear: To study the effects of UV-B radiation on rice grain morphology, proanthocyanidin content, distribution and synthesis, and analyze the changes of antioxidant capacity of red rice by feeding aging mice.

Abstract is not clear and should be improved!

Write the whole words of LAR for the first time.

Introduction

L 37-38 Write full stop after UV-B radiation” The accumulation of secondary metabolites is a marker process of plant response to UV-B radiation, For example, phenolic compounds (especially flavonoids), alkaloids and terpenoids.

The section "Introduction" contains enough information, the literature is adequate. English must be checked! Please put spaces where they should be!

Materials and Methods

L 455 room temperature [56,57].Preparation of control solution: accu..Write  space before “Preparation…!

L 470: ,μg·mL-1)is the abscissa and the peak area (Y) is the ordinate,… Write space after parenthesis

The style of writing varies a lot within a section! It is necessary to unify it! Past and present times are mixed.

Results

L 207 Table 2. This is a table. Tables should be placed in the main text near to the first time they are cited. ??????

Explanatory data is missing under the table 2

Legend is missing under the table 3

The results are interesting. Please correct your English.

Discussion

In some places it is difficult to follow the discussion because the English is not good enough. I suggest you improve your English!

Conclusion:

Write at least three sentence.

Specific comments

The authors present a study of high practical relevance. I found your study very interesting and informative.

My suggestions: major revision

Reviewer 3 Report

Dear Authors, thank you for the possibility to contribute to your work as a reviewer. This study aims to investigate the on-field effect of increased UV-B radiation on the proanthocyanidin content of red rice. I think that the experiment is well planned and recorded. The introduction provides a good insight into the topic, and the discussion is a very strong part of the ms.

I suggest, however, to reword certain parts of the materials and methods, specifically those that describe the methods as a recipe; it should rather be a report. Instead of saying: install a lamp rack with …, say: a lamp rack was installed…, and so on.

Author Response

Dear Authors, thank you for the possibility to contribute to your work as a reviewer. This study aims to investigate the on-field effect of increased UV-B radiation on the proanthocyanidin content of red rice. I think that the experiment is well planned and recorded. The introduction provides a good insight into the topic, and the discussion is a very strong part of the ms.

I suggest, however, to reword certain parts of the materials and methods, specifically those that describe the methods as a recipe; it should rather be a report. Instead of saying: install a lamp rack with …, say: a lamp rack was installed…, and so on.

Response: Thank you for your advice. We have completely revised the materials and methods.

For example: UV-B radiation treatment: a lamp rack was installed with adjustable length above each row of rice, UV-B lamp tube was erected (40 W, wavelength 280 nm-320 nm, provided by Shanghai Gucun instrument company). The radiation intensity was measured at the top of rice plant with UV-B radiation meter (provided by photoelectric instrument factory of Beijing Normal University). Four groups of natural light (0 kJ·m-2·d-1) and UV-B radiation (2.5, 5.0, 7.5 kJ·m-2·d-1) were set up, which were equivalent to local ozone attenuation of 0%, 10%, 20%, 30% respectively. The daily irradiation time was 7 h (10:00-17:00), and UV-B radiation treatment was not carried out in cloudy or rainy days.

Round 2

Reviewer 1 Report

No comments 

Author Response

We have revised the language of the article and adjusted the content of the article to make it more readable.

Reviewer 2 Report

In the manuscript Enhanced UV-B radiation induced the proanthocyanidins accumulation in red rice grain of traditional rice cultivars and increased antioxidant capacity in aging mice, authors Xiang Li, Jianjun Sheng, Zuran Li, Yongmei He, Yanqun Zu and Yuan Li,  hypothesized that UV-B radiation induced proanthocyanidins synthesis in red rice to improve its antioxidant capacity.

Abstract

Firs statement in the Abstract section is not clear and English is also not OK!

Introduction

L 39 …visible light and photosynthetic effective radiation... How they differ?

L 50-60. Decide will you use past tense or present tense in the Introduction section!

L 75-79 Proanthocyanidins are water-soluble, non-toxic, non-allergic and other characteristics, and are natural free radical scaveners and antioxidants. What does it mean “other characteristics”?

L 81 [20].The… Write space!

L87 …Oxygen free radicals, such as superoxide anion and light radical free radicals,… What are light radical free radicals?

L 101 aging [28].Malondi… write space!

L 105- 106 The following statement is not clear: It is also important to reduce the level of oxidative stress and maintain the balance of redox for people's health and longevity.

L 113 n [32].At pr… write space

L 123 The following part is not clear: and anti-aging ability of red rice to the enhancement of UV-B radiation.

Materials and Methods

L 382 The following statement is unclear: The local soil is hydroponic artificial soil, …

L 411 The following statement is unclear: The air was extracted to sink grain. It was fixed at room temperature for 2 h and transfered to 4 °C for storage.

L 414-415 The following statements are not clear: The rear stationary solution was sucked off. 0.1mol·L-1 phosphoric acid buffer was added to soak for 3 times for 10-15 minutes each time.

L 417 The following statement is unclear: 100% ethanol for 3 times, and the resi- dence time of each stage is 10 min.

L 419 T The following statement is unclear: Then put it into the critical point dryer (K850, UK) for drying.

L 432 The following statement is not clear Then 50 times with 80% methanol water was diluted to detect proanthocyanidin C1

L 436-440 The following statement is not clear: Accurately take 20.0 μ L of standard solution (100 μ g·mL-1), dilute it with 80% methanol water to a constant volume of 1.00 mL to prepare a standard solution with a concentration of 2000 ng·mL-1, and dilute it with 80% methanol water to prepare a standard series solution with a concentration of 2000, 1000, 500, 200, 100, 50.0, 20.0, 10.0, 5.00, 2.00, 1.00, 0.50, 0.20 and 0.10 ng·mL-1.

L 446-452 The following statements are not clear: Mobile phase composition: A-water (0.1% formic acid), B-acetonitrile, running time: 8 min, sample volume: 4 μ L, sample gradient elution procedure was shown in Table 4. Mass spectrum conditions: ion source: ESI ion source, curtain gas: 20 arb, collision gas: 9 arb, ion spray voltage: - 4500V, ion source temperature: 450 ° C, ion source gas1: 55 arb, ion source gas2: 55 arb, MRM acquisition parameters: according to the above and the chromatographic and mass spectrum conditions in Table 4.

Check spaces in Table 4!

L 465-466 The following statement is not clear: 20 μL supernatant was added with 780 μL 0.1 mol·L-1 boric acid buffer and 200 μL 0.02 mol·L-1 phenyl- alanine.

L 475 English is not clear: was added with 50 mmol Tris-HCl

L 476-477 English is not Ok in the following statement: Take the crude enzyme solution and add 50 μL 1mg·mL-1 chalcone was bathed in 34 °C water for 30min, and the same amount of crude enzyme solution was boiled in water for 10 min as the control.

L 487 English is not Ok in the following statement. Then it was extracted with 0.2 mL distilled water for three times.

L 491-492 English is not Ok in the following statement: An enzyme activity unit is expressed the amount of enzyme required to decompose dihydroquercetin per milligram protein per minute to produce 1mmol catechin at 30 °C, pH=7.5.

L 506-512 English is not Ok in the following statement: When the rice was mature, the rice on the ear was separated from the ear stalk. The rice was crushed into powder with a grinder. It was mixed with the crushed basic feed (provided by the De- partment of Experimental Zoology of Kunming Medical University, which is mainly com- posed of wheat bran, soybean flour and corn flour) in 1:1 ratio to ensure the daily nutrition intake of mice. The mixed feed was pressed into strips. The red rice with different inten- sities of UV-B radiation was prepared as food for mice. After an acclimatizing period of 5 days, the mice were randomly divided into seven groups

L 513 English is not Ok in the following statement: They were feeding general mice

L 544 English is not Ok in the following statement: The liver, brain and heart of mice were made…

L 561 English is not Ok in the following statement: Use microseoft Excel to sort out the data and calculate the mean and variance.

The style of writing varies a lot within a section! It is necessary to unify it! Past and present times are mixed. English should be improved!!

Results

L143 pe(Figure 1)… Space!

L 188 ging model feeding general food group Font?

Figure 5. It is written rice and red rice.

Please correct your English.

Discussion

L252 What is plant light morphology?

L265-265 The following statement is unclear: Nutrient transport of rice endosperm is to transport the grouting substances in the apoplast to the endosperm through aleurone layer [42].

L 314: English is not Ok in the following statement: Proanthocyanidins synthesis precursors such as epi- catechin are synthesized

L 365English is not Ok in the following statement And found that the expression of SOD and catalase increased.

I suggest you improve your English!

Conclusion:

Write a more fluent text

Specific comments

English needs to be better throughout the text. Spaces are missing somewhere.

 My suggestions: minor revision

Author Response

Abstract

Firs statement in the Abstract section is not clear and English is also not OK!

Response: Modified as:Proanthocyanidins are major UV-absorbing compounds.

Introduction

L 39 …visible light and photosynthetic effective radiation... How they differ?

Response: We deleted “and photosynthetic effective radiation”.

L 50-60. Decide will you use past tense or present tense in the Introduction section!

Response: We will decide to use the present tense uniformly

L 75-79 Proanthocyanidins are water-soluble, non-toxic, non-allergic and other characteristics, and are natural free radical scaveners and antioxidants. What does it mean “other characteristics”?

Response: The original idea is that there are other characteristics, such as ease of handling. We forgot the requirements for accurate expression of scientific research papers, so we deleted “other”.

L 81 [20].The… Write space!

Response: Done.

L87 …Oxygen free radicals, such as superoxide anion and light radical free radicals,… What are light radical free radicals?

Response: Modified as: hydroxyl radical.

L 101 aging [28].Malondi… write space!

Response: Done.

L 105- 106 The following statement is not clear: It is also important to reduce the level of oxidative stress and maintain the balance of redox for people's health and longevity.

Response: Modified as: Reducing the level of oxidative stress and maintaining the redox balance contribute to health and longevity.

L 113 n [32].At pr… write space

Response: Done.

L 123 The following part is not clear: and anti-aging ability of red rice to the enhancement of UV-B radiation.

Response: We split this sentence into two parts. Modified as: analyzed the response of the morphology, proanthocyanidins content and proanthocyanidins synthesis of red rice to the enhancement of UV-B radiation. The effects of UV-B radiation on the antioxidant capacity of rice were evaluated by feeding aging model mice.

Materials and Methods

L 382 The following statement is unclear: The local soil is hydroponic artificial soil, …

Response: We don't know the professional terms accurately. Thank you for pointing out. Modified as: hydragric anthrosols

L 411 The following statement is unclear: The air was extracted to sink grain. It was fixed at room temperature for 2 h and transfered to 4 °C for storage.

Response:  Modified as: The grain was fixed in 2% glutaraldehyde fixed solution (prepared with 0.1 mol·L-1, pH=7.2 sodium phosphate buffer) for 2 h.

L 414-415 The following statements are not clear: The rear stationary solution was sucked off. 0.1mol·L-1 phosphoric acid buffer was added to soak for 3 times for 10-15 minutes each time.

Response:  Modified as: The fixed seeds were rinsed again with pH=7.4 0.1mol·L-1 phosphoric acid buffer for 3 times.

L 417 The following statement is unclear: 100% ethanol for 3 times, and the resi- dence time of each stage is 10 min.

Response:  Modified as: Dehydrate 3 times with 100% ethanol, and the dehydration process time of each stage was 10 minutes.

L 419 T The following statement is unclear: Then put it into the critical point dryer (K850, UK) for drying.

Response:  Modified as: After the gradual dehydration and critical point drying by the critical point dryer (K850, UK),

L 432 The following statement is not clear Then 50 times with 80% methanol water was diluted to detect proanthocyanidin C1

Response:  Modified as: Because the content of proanthocyanidin C1 exceeded the detection range, it needed to be diluted by 50 times with 80% methanol water before determining.

L 436-440 The following statement is not clear: Accurately take 20.0 μ L of standard solution (100 μ g·mL-1), dilute it with 80% methanol water to a constant volume of 1.00 mL to prepare a standard solution with a concentration of 2000 ng·mL-1, and dilute it with 80% methanol water to prepare a standard series solution with a concentration of 2000, 1000, 500, 200, 100, 50.0, 20.0, 10.0, 5.00, 2.00, 1.00, 0.50, 0.20 and 0.10 ng·mL-1.

Response:  Modified as: 20.0 µL of standard solution (100 µg·mL-1) was accurately taken.It was diluted with 80% methanol water to prepare a standard series solution with a concentration of 2000, 1000, 500, 200, 100, 50.0, 20.0, 10.0, 5.00, 2.00, 1.00, 0.50, 0.20 and 0.10 ng·mL-1.

L 446-452 The following statements are not clear: Mobile phase composition: A-water (0.1% formic acid), B-acetonitrile, running time: 8 min, sample volume: 4 μ L, sample gradient elution procedure was shown in Table 4. Mass spectrum conditions: ion source: ESI ion source, curtain gas: 20 arb, collision gas: 9 arb, ion spray voltage: - 4500V, ion source temperature: 450 ° C, ion source gas1: 55 arb, ion source gas2: 55 arb, MRM acquisition parameters: according to the above and the chromatographic and mass spectrum conditions in Table 4.

Check spaces in Table 4!

Response:  The spaces in Table 4 and Table 5 have been removed. Modified as: The following mobile phase was used: A: water containing 0.1% formic acid; B: 100% ace-tonitrile. 4 µL sample volumes were taken gradient elution for 8 min. Sample gradient elu-tion procedure was shown in Table 4.

The optimum operating parameters were determined by electro spray ionization (ESI) interface in positive ion mode. A generic mass spectrometry parameters of the analyte were developed and used for the analysis. These parameters were: curtain gas: 20 arb; col-lision gas: 9 arb; ion spray voltage: - 4500V; ion source temperature: 450 ° C; ion source gas 1: 55 arb; ion source gas 2: 55 arb; multiple reaction monitoring (MRM) acquisition pa-rameters: according to the above and the chromatographic and mass spectrum conditions in Table 4.

L 465-466 The following statement is not clear: 20 μL supernatant was added with 780 μL 0.1 mol·L-1 boric acid buffer and 200 μL 0.02 mol·L-1 phenyl- alanine.

Response:  Modified as: 20 μL crude enzyme solution was added with 780 μL 0.1 mol·L-1 boric acid buffer and 200 μL 0.02 mol·L-1 phenylalanine.

L 475 English is not clear: was added with 50 mmol Tris-HCl

Response:  Modified as: 0.75 mL of crude enzyme solution was diluted to 2mL by 50 mmol Tris-HCl (containing 50 mM Tris HCl, 50 mM KCN, pH=7.4).

L 476-477 English is not Ok in the following statement: Take the crude enzyme solution and add 50 μL 1mg·mL-1 chalcone was bathed in 34 °C water for 30min, and the same amount of crude enzyme solution was boiled in water for 10 min as the control.

Response:  Modified as: 0.75 mL of crude enzyme solution was diluted to 2mL by 50 mmol Tris-HCl (containing 50 mM Tris HCl, 50 mM KCN, pH=7.4). 50 μL 1mg·mL-1 chalcone was added and bathed in 34 ℃ water for 30min. The same amount of crude enzyme solution was boiled in water for 10 min as the control.

L 487 English is not Ok in the following statement. Then it was extracted with 0.2 mL distilled water for three times.

Response:  Modified as: The residue was washed with distilled water (three times with a total of 0.6 mL of distilled water) to remove ethyl acetate.

L 491-492 English is not Ok in the following statement: An enzyme activity unit is expressed the amount of enzyme required to decompose dihydroquercetin per milligram protein per minute to produce 1mmol catechin at 30 °C, pH=7.5.

Response:  Modified as: DFR enzyme activity was expressed as the amount of enzyme required to decompose dihydroquercetin to produce 1 mmol catechin per milligram of protein per minute at 30 ℃, pH=7.5.

L 506-512 English is not Ok in the following statement: When the rice was mature, the rice on the ear was separated from the ear stalk. The rice was crushed into powder with a grinder. It was mixed with the crushed basic feed (provided by the De- partment of Experimental Zoology of Kunming Medical University, which is mainly com- posed of wheat bran, soybean flour and corn flour) in 1:1 ratio to ensure the daily nutrition intake of mice. The mixed feed was pressed into strips. The red rice with different inten- sities of UV-B radiation was prepared as food for mice. After an acclimatizing period of 5 days, the mice were randomly divided into seven groups

Response:  Modified as: Rice grains were crushed into powder with a grinder when rice was mature. It was mixed with crushed basic feed in a ratio of 1:1 to ensure the daily nutrition intake of mice The basic feed was provided by the Department of Experimental Zoology of Kunming Medical University, which is mainly composed of wheat bran, soybean flour and corn flour. The mixed mice diet of red rice treated with different intensities of UV-B radiation was prepared. After an acclimatizing period of 5 days, the mice were randomly divided into seven groups.

L 513 English is not Ok in the following statement: They were feeding general mice

Response:  Modified as: The mice were feeding basic feed group

L 544 English is not Ok in the following statement: The liver, brain and heart of mice were made…

Response:  Modified as: The mice liver, brain and heart were homogenized into 10% tissue homogenate with 4 ℃ normal saline.

L 561 English is not Ok in the following statement: Use microseoft Excel to sort out the data and calculate the mean and variance.

Response:  Modified as: The data was sorted out by Microsoft Excel and calculated the mean and variance.

The style of writing varies a lot within a section! It is necessary to unify it! Past and present times are mixed. English should be improved!!

Results

L143 pe(Figure 1)… Space!

Response:  Done.

L 188 ging model feeding general food group Font?

Response:  Done.

Figure 5. It is written rice and red rice.

Response:  Done.

Please correct your English.

Discussion

L252 What is plant light morphology?

Response:  Modified as: photomorphogenesis

L265-265 The following statement is unclear: Nutrient transport of rice endosperm is to transport the grouting substances in the apoplast to the endosperm through aleurone layer [42].

Response:  Modified as: The filling material in the exoplast is transported to the endosperm through the aleurone layer

L 314: English is not Ok in the following statement: Proanthocyanidins synthesis precursors such as epi- catechin are synthesized

Response:  Modified as: Synthesis precursors of proanthocyanidins such as epicatechin are synthesized

L 365English is not Ok in the following statement And found that the expression of SOD and catalase increased.

Response:  Modified as: And the expression of SOD and catalase were increased.

I suggest you improve your English!

Conclusion:

Write a more fluent text

Response:  Modified as: UV-B radiation induces the synthesis of proanthocyanidins B2 and C1 in red rice. The neuronal damage in hippocampus of aging model mice was improved by red rice. The antioxidant capacity of red rice was significantly improved by proanthocyanidins synthesis induced by 5.0 kJ·m-2·d-1 UV-B radiation.

Specific comments

English needs to be better throughout the text. Spaces are missing somewhere.

Response: Thank you for your suggestion. We plan to polish the article in English again.

 My suggestions: minor revision